Partial revision of the neustonic genus Scapholeberis Schoedler, 1858 (Crustacea: Cladocera): decoding of the barcoding results

Garibian Petr G. 1
Neretina Anna N. 1
Taylor Derek J. 2
http://orcid.org/0000-0002-8863-6438 Kotov Alexey A. 1 alexey-a-kotov@yandex.ru
1 A.N. Severtsov Institute of Ecology and Evolution, Russian Academy of Sciences , Moscow , Russia
2 Department of Biological Sciences, State University of New York at Buffalo , Buffalo, NY , USA
Poyarkov Nikolay
Electronic publication date: 2020 Nov 25
Publication date: 2020
Volume: 8
Electronic Location ID: e10410
Received 2020 Jul 13; Accepted 2020 Nov 2
Copyright: © 2020 Garibian et al.
Copyright year: 2020
Copyright holder: Garibian et al.
License: This is an open access article distributed under the terms of the Creative Commons Attribution License, which permits unrestricted use, distribution, reproduction and adaptation in any medium and for any purpose provided that it is properly attributed. For attribution, the original author(s), title, publication source (PeerJ) and either DOI or URL of the article must be cited.
License URL: https://creativecommons.org/licenses/by/4.0/

Keywords: Scapholeberis, Morphology, Genetics, Barcoding, New species, Integrative taxonomy, Biogeography

Funding: National Institute of Biological Resources (NIBR) Ministry of Environment (MOE) of the Republic of Korea (NIBR201701201) Joint Ethiopian-Russian Biological Expedition (JERBE) Russian Science Foundation 18-14-00325 Sampling in South Korea and provisory examination of the specimens from this country are supported by a grant from of the National Institute of Biological Resources (NIBR), funded by the Ministry of Environment (MOE) of the Republic of Korea (NIBR201701201). Sampling in Ethiopia was supported by the Joint Ethiopian-Russian Biological Expedition (JERBE). The morphological and genetic study is supported by the Russian Science Foundation (project No. 18-14-00325). The funders had no role in study design, data collection and analysis, decision to publish, or preparation of the manuscript.

==============================
Water fleas (Crustacea: Cladocera) are among the most intensively studied freshwater invertebrates. However, ecologically important daphniids that live on the surface layer (neuston) remain taxonomically confused. Here we attempt to reconcile genetic and morphological information for the neustonic genus Scapholeberis Schoedler, 1858 (Cladocera: Daphniidae) and present the first revision of the Scapholeberis kingii species group. We analyzed new and existing mitochondrial DNA sequences (сytochrome C oxidase subunit I gene region) together with morphology for all but one of the known species of the neustonic daphniids. Morphological comparisons of available populations, belonging to the Scapholeberis kingii species group from several Australian, Asian and African localities, revealed, that they are almost identical according to parthenogenetic females. However, Australian populations can be reliably distinguished from Asian ones based on the morphology of gamogenetic females. Mitochondrial DNA data analyses revealed divergent lineages (>17% for the DNA barcoding COI region) for the three different species (Australia, Asia and Africa). Based on this set of data, we redescribed S. kingii Sars, 1888 from Australia, its terra typica, and described a new species, S. smirnovi sp.nov. from the Russian Far East, Korea and Japan. The status of populations from Ethiopia and the Republic of South Africa remained unclear, because in the African material and the putative type material, we found only parthenogenetic females. Our results provide an integrative revision of the S. kingii species group and improve the taxonomic scaffold used for barcoding and genomics for the remaining species groups in the daphniid genus Scapholeberis.

Introduction

Integrative taxonomy combines the evidence from disparate biological disciplines to better understand biodiversity. This approach has been particularly fruitful for taxonomically challenging yet well-studied aquatic groups such as the water fleas (Crustacea: Branchiopoda: Cladocera). For some cladoceran taxa successful advances have been made by morphological (Smirnov, 1992, 1996; Van Damme, Sinev & Dumont, 2011; Neretina & Sinev, 2016) or genetic evidence alone (Adamowicz et al., 2009; Bekker et al., 2016; Thielsch et al., 2017). For some problematic cladoceran taxa, a combination of approaches has resulted in taxonomic progress (Belyaeva & Taylor, 2009; Kotov, Ishida & Taylor, 2009; Quiroz-Vázquez & Elías-Gutiérrez, 2009; Sinev, Karabanov & Kotov, 2020). The integrative approach has been particularly useful for taxa that lack distinguishing characters for parthogenetic females. For cladocerans, the sexual stages appear sporadically, but can be a rich source of diagnostic morphological characters (see review in Kotov, 2013). Genetic approaches, such as formal genetic barcoding (Hebert et al., 2003), have much value for the discovery of novel lineages and taxonomic diagnoses. However, taxonomic advances with genetic information alone are problematic because the existing taxonomic scaffold (i.e., from the 19th of 18th centuries) is based on morphology (Kotov & Gololobova, 2016; Dupérré, 2020). Moreover, as museum samples, including type materials, are generally not amenable to genetic study (but see Umetsu et al., 2002; Turko et al., 2019), taxonomic advances are often limited to morphological evidence.

At the same time, genetic data (sequences of different genes) for cladocerans (as well as other organisms) from different geographic regions are rapidly accumulating in specialized databases such as GenBank (Benson et al., 2012). A massive accumulation of cytochrome C oxidase subunit I sequences (COI data) is available from the Barcoding of Life initiative (Hebert et al., 2003). The coordination of this genetic information with formal taxonomic knowledge, even with the modest aim of accurate species identifications, is a considerable challenge.

The aim of the present paper is to apply the integrative approach to the taxonomic problems of cladocerans associated with the surface layer of standing waters, with a focus on the genus Scapholeberis Schödler, 1858 (Anomopoda: Daphniidae: Scapholeberinae). Since the revision of Dumont & Pensaert (1983), most efforts to understand the diversity within this genus have been local (Hudec, 1983; Elmoor-Loureiro, 2000; Elías-Gutiérrez et al., 2008; Quiroz-Vázquez & Elías-Gutiérrez, 2009; Hudec, 2010; Kotov, Jeong & Lee, 2012; Andrade-Sossa, Buitron-Caicedo & Elías-Gutiérrez, 2020). Recently, a global phylogenetic study of the subfamily based of 402 multigene sequences from the 12S rRNA, 16S rRNA, and tRNA (val) regions of the mitochondrial genomes was carried out (Taylor, Connelly & Kotov, 2020). This study revealed an unexpectedly high lineage diversity in the Eastern Palearctic. Other regions, such as Africa, remained unexamined according to current standards of cladoceran taxonomy. Notably, the within-genus divergences for neustonic taxa were much greater than that found within other daphniid genera (Taylor, Connelly & Kotov, 2020). We were unable to reconcile the newly uncovered taxa with existing databases, genome projects, and taxonomy or to assess if the marked divergences were limited to non-protein coding regions. Here we address some geographic sampling gaps (such as Africa), attempt to unify the genetic (including DNA barcoding and genome projects) and morphological knowledge, and revise the taxonomy of the genus Scapholeberis. We collect new COI sequences and revise the taxonomy of the widespread and historically confused Scapholeberis kingii Sars, 1888 species group using an integrated approach.

Materials and Methods

Collecting samples and their preliminary analysis

Numerous samples from different localities in different continents were collected by our team or by our colleagues via small-sized plankton nets (with mesh size 50 µm) and fixed with 4% formaldehyde or 96% ethanol in the fields, immediately after sampling. Sampling in non-protected water bodies of Russia does not require special permissions. Sampling in South Korea was conducted in frames of the program of the National Institute of Biological Resources (NIBR), of the Republic of Korea. Sampling in Ethiopia was conducted in frames of work of the Joint Ethiopian-Russian Biological Expedition (JERBE), with permission from the Ministry of Environment of Ethiopia to JERBE. Samples from Australia were obtained from colleagues having appropriate permissions.

All samples were preliminarily examined using a stereoscopic microscope. Individuals of Scapholeberis in them were initially identified via available references only according to morphological features (mainly, shape of head and rostrum from the ventral view) (Dumont & Pensaert, 1983; Kotov et al., 2010).

Genetics

Before genetic analysis, identification of each parthenogenetic female was re-checked under a binocular stereoscopic microscope in order to avoid mistakes, because some samples contained several Scapholeberis species. Selected individuals were placed into 96-well PCR plates and dried from ethanol on air. DNA of single individuals was extracted using DNA QuickExtract (Epicenter) as modified by Ishida, Kotov & Taylor (2006). PCR reactions were carried out in 25 µL or 50 µL volumes using the Promega GoTaq Master mix protocol with 5 µL of DNA extraction using HCO/LCO primers of Folmer et al. (1994). PCR cycling conditions were 95 °C for 2 m, 95 °C for 30 s, 48 °C for 30 s and 72 °C for 1 m for 39 cycles, followed by 72 °C for 5 m. The sizes of the PCR products were verified by agarose gel electrophoresis. PCR products were then purified and sequenced by TACGEN (California). Amplicons were sequenced in both directions and the contigs were assembled in Geneious R7. The authenticity of newly obtained sequences was verified by BLAST comparisons. Additional sequences were obtained from NCBI GenBank. The alignment was carried out in the online version of MAFFT 7 using the default settings. Phylogenetic trees were estimated using a Maximum Likelihood (ML) optimality criterion (with a GTR+I+gamma model) and the Subtree Pruning and Regrafting branch-swapping algorithm in Seaview 4.7. Violin plots (which show the full distribution of the data) were created in R for major taxa based on pairwise Kimura’s 2-parameter distances (also calculated in Seaview). Branch support for the ML tree was estimated by the transfer bootstrap expectation method (using BOOSTER: https://booster.pasteur.fr/) which typically shows less “false” erosion of support compared to nonparametric bootstrap for deeper nodes (Lemoine et al., 2018). Bayesian analyses (BI) were performed in MrBayes v.3.2.6 (Ronquist et al., 2012). Four independent Markov chain Monte Carlo (MCMC) analyses were run simultaneously for 100,000 generations and sampled every 500 generations. The site rate parameter (rates) was gamma plus invariable sites (invgamma) and the number of substitution types (nst) was six. The first 25% of the generations were discarded as the burn-in. Phylograms were visualized using the FigTree Version 1.4.4. The ML tree was rooted using specimens of the genus Megafenestra as outgroups.

Original sequences are deposited to the Genbank under accession numbers MT371605–MT371659.

Morphological analysis

The morphology of populations from Australia and Asia (southern part of the Russian Far East and South Korea), containing both parthenogenetic and ephippial females, was examined in detail with the aim of finding diagnostic characters. Only parthenogenetic females from Ethiopia and the Republic of South Africa were examined because ephippial females and males were lacking. Specimens of Scapholeberis from presorted samples were selected under a binocular stereoscopic microscope LOMO, and then studied in toto under optical microscopes Olympus BX41 or Olympus CХ 41 in a drop of glycerol-formaldehyde or a glycerol-ethanol mixture. Then at least two parthenogenetic females and two ephippial females (if available) from each locality were dissected under a stereoscopic microscope for the study of appendages and postabdomen. Drawings were prepared via a camera lucida attached to optical microscopes. Several individuals from each population were dehydrated in a series of ethanol washes (30%, 50%, 70% and 95%) and 100% acetone and then dried using hexamethyldisilazane (Laforsch & Tollrian, 2000). Dried specimens were mounted on aluminium stubs, coated with gold in a S150A Sputter Coater (Edwards, UK), and examined under a scanning electron microscope (Vega 3 Tescan Scanning Electron Microscope, TESCAN, Czech Republic). We used a system of setae enumeration outlined by Kotov (2013).

In cases of dubious homologies, the numbers are supplied by question marks.

Nomenclatural acts

The electronic version of this article in Portable Document Format will represent a published work according to the International Commission on Zoological Nomenclature (ICZN), and hence the new names contained in the electronic version are effectively published under that Code from the electronic edition alone. This published work and the nomenclatural acts it contains have been registered in ZooBank, the online registration system for the ICZN. The ZooBank Life Science Identifiers (LSIDs) can be resolved and the associated information viewed through any standard web browser by appending the LSID to the prefix http://zoobank.org/. The LSID for this publication is: urn:lsid:zoobank.org:pub:A4A3415D-857E-42E5-9103-B8D48AC60832. The online version of this work is archived and available from the following digital repositories: PeerJ, PubMed Central and CLOCKSS.

Results

COI Phylogeny

A total of 106 Scapholeberine sequences (58 from this study) were aligned and analyzed. We detected 21 main mitochondrial clades of Scapholeberinae (Figs. 1 and 2; Table S1). We used the clade labels proposed by Taylor, Connelly & Kotov (2020). Lineages novel to the present study are labelled: X, Y, L2, J1–J4. Deep branches within Scapholeberis had low to moderate support in the ML tree. In contrast, the differentiation of terminal taxa (species) was well-supported, as was the separation of major morphologically-based species/species groups: S. mucronata (clades A–C and X, green in Fig. 2), S. rammneri (clades F–H and Y, red), S. freyi (clades J1–J4, black), S. kingii (clades K, L1, L2, grey), S. spinifera (clade M), and S. cf. microcephala- armata (clades E and N) (Figs. 1 and 2).

Figure 1 Map of populations of Scapholeberis and Megafenestra studied here genetically.

Symbols correspond to mitochondrial clades (see Figure 2): (A) populations of the S. mucronata species group (northern hemisphere); (B) populations of the S. rammneri species group in the northern hemisphere; (C) populations of the S. freyi species group (western hemisphere); (D) populations of Megafenestra (clear symbols), S. microcephala, S. smirnovi sp.nov., S. armata, S. cf. microcephala in northern hemisphere; (E) all populations revealed in southern hemisphere. The base maps are from the public domain atlas in the desktop app, Marble 2.2.20 (http://edu.kde.org/marble). Symbols were placed manually in Microsoft PowerPoint using the output from DIVA-GIS 7.5 (https://www.diva-gis.org/) as a guide. Note that the base maps and symbols are basically same as in Taylor, Connelly & Kotov (2020), but just the only localities are represented from where the COI sequences were obtained here in addition to Taylor, Connelly & Kotov (2020).

Figure 2 Maximum likelihood mitochondrial phylogeny of neustonic daphniids (Scapholeberis and Megafenestra).

Bold letters (A–Q and X–Y) indicate geographic clades. Numbers at the nodes indicate Bayesian posterior probabilities and Transfer Bootstrap Expectations (TBE). Colours represent major species groups in the Scapholeberinae: Scapholeberis mucronata group (green), S. rammneri group (red), S. freyi group (black), S. kingii group (grey), S. armata-microcephala clade (blue), genus Megafenestra (white). The tree is outgroup-rooted using sequences from the genus Megafenestra. See Table S1 for individual sequences.

Table 1 Differentiation of presently recognised species of the Scapholeberis kingii group in Eurasia based on morphological characters.

Taxon	S. kingii	S. intermedius	S. smirnovi sp.nov.	
Rate of distance between the center of ocellus and eye to distance from the center of ocellus to the tip of rostrum	Almost 2	Almost 3	About 5	
On thoracic limb I, the ratio between seta 1′ and seta 2	Almost 2.5 (i.e., seta 2 is relatively short)	Almost 1.5 (i.e., seta 2 is relatively long)	Almost 2.5 (i.e., seta 2 is relatively short)	
In ephippial females, area between two keels of ephippium	Strongly elongated, keels not projected laterally out of body dorsal contour	Unknown	Strongly rounded, keels strongly projected laterally out of body dorsal contour	

The S. mucronata species group (Figs. 1 and 2) had four main geographic clades (A+B+C+X). Clade A (S. mucronata s. str.) was detected only in Western and Central Europe; clade B was detected from European Russia to Yakutia and Alaska; clade C was found in Western Alaska only. Clade X was detected in the vicinity of Churchill, Manitoba (Jeffery, Elías-Gutiérrez & Adamowicz, 2011).

The S. rammneri species group (Figs. 1 and 2) had five main geographic clades (F+G+H+I+Y). Clade F (S. rammneri s.str.) was found in a single locality in Mongolia; clade G was present in two localities in Eastern Siberia; clade H was widely distributed in North America and in a single locality in Patagonia; clade I was detected only in a single locality in Patagonia. Clade Y was found in a single locality in Israel.

The S. freyi species group (Figs. 1 and 2) was represented by four main clades (J1–J4). Clade J1 (S. freyi s.str.) was detected in many localities in North America; clade J2 (S. duranguensis) was found in two localities in Mexico; clade J3 was found in three localities on the Yucatan Peninsula (sequences of Elías-Gutiérrez et al., 2008 and Prosser, Martínez-Arce & Elías-Gutiérrez, 2013); clade J4 was present in a single locality in Brazil (sequence directly submitted to the GenBank and then described as S. yahuarcaquensis by Andrade-Sossa, Buitron-Caicedo & Elías-Gutiérrez (2020) and found also in Amazon basin in Colombia) and a single locality in Belgium (also a direct submission to the GenBank).

The S. microcephala-armata species group (Figs. 1 and 2) was represented by two main clades, E from Alaska and Far east, and N from North America.

The S. kingii species group (Figs. 1 and 2) was represented by three clades (K, L1 and L2). Clade K (Scapholeberis kingii s. str.) was detected only in Australia; clade L1 was found in Japan and China; clade L2 was found in a single locality in Ethiopia.

The genus Megafenestra (Figs. 1 and 2) was represented by three clades: clade O (M. aurita s.str.) was found in Europe (Ukraine and Switzerland), clade P was present in Alaska only; clade Q (M. nasuta s.str.) was present in New York State, USA.

Sequence pairs within each genus (Megafenestra and Scapholeberis) had maximum K2 parameter distances that exceeded 30% (Fig. 3). Indeed, the mean pairwise sequence divergence within Scapholeberis exceeded 20%. Notably, pairwise sequence divergences within each major species group exceeded 20% as well. The closest members of the S. kingii complex were from Japan and Africa with a 17.4% distance estimate (Fig. 2).The large divergences within genera in DNA sequences were not accompanied by divergences in COI amino acid sequences. The most common protein sequence for example was >99% similar to that found in the genus Daphnia (e.g., AAL08864.1). Synapomorphic amino acid substitutions in Scapholeberis included: a glycine to an alanine for S. kingii (Australia), an alanine to a serine for the S. mucronata group, and a serine to an alanine in S. microcephala.

Figure 3 Violin plots of pairwise Kimura’s 2 Parameter Distances from the COI region of mitochondrial DNA in clades of neustonic daphniids (Scapholebeberinae).

Horizontal bars indicate means. Gray rectangles show the ranges. Taxa are genera or species groups in the Scapholeberinae. See Table S1 for individual sequences.

Morphological analysis

Order Anomopoda Sars, 1865

Family Daphniidae Straus, 1820

Subfamily Scapholeberinae Dumont & Pensaert, 1983

Genus Scapholeberis Schödler, 1858

Scapholeberis kingii species group

Diagnosis. Species of medium size for the genus (length of adult parthenogenetic female up to 0.75 mm without mucro). Body with typical features of the genus (see Dumont & Pensaert, 1983), relatively elongated. In lateral view, head relatively large, without keel. Rostrum relatively short and blunt. In ventral view posteroventral portion of head forms a three-lobed rostrum, due to a shallow depression at the insertion point of antenna I on each side, its middle lobe rounded, with minute frontal head pore. Dorsal head pores absent. Head and valves without short denticles, spines or protuberances. Ventral margin of valve straight. Posteroventral angle with short mucro. Adhesive ventral rim of valves modified into “sucker-plate” (in terms of Dumont & Pensaert, 1983), no setae along most part of the sucker length except few rarely located setae at anteriormost portion and several sparsely located setae at posterior portion near mucro. Inner surface of posterior margin with broad “hyaline membrane” extending posterior rim and “denticulated membrane” consisting of row of short setules along posterior rim. Five pairs of thoracic limbs, proportions between seta 1′ and seta 2 of thoracic limb I are important for species identification. Ephippium with single egg and two longitudinal keels.

Differentiation of species is based on characters listed in Table 1.

1. Scapholeberis kingii Sars, 1888

Figures. 4–9

Figure 4 Scapholeberis kingii Sars, 1888, parthenogenetic and ephippial females from Farm Dam, New South Wales, Australia.

A–D, Adult parthenogenetic females; E, Juvenile parthenogenetic female; F–H, Ephippial females. A, Parthenogenetic female, lateral view. B, Adult parthenogenetic female, dorsal view. C, Head, ventral view. D, Labrum. E, Juvenile parthenogenetic female, lateral view. F, Ephippial female, lateral view. G, Ephippial female, dorsal view. H, Ornamentation of ephippium. Scale bars = 0.1 mm.

Figure 5 Scapholeberis kingii Sars, 1888, parthenogenetic females from Farm Dam, New South Wales, Australia.

(A) Valve, ventral view. (B) and (C) Armature of valve. (D) and (E) Posteroventral portion of valve, inner view. (F) and (G), Postabdomen. (H) and (I), Postabdominal claw. (J) and (K), Antenna I. (L) and (M), Antenna II. Scale bars = 0.1 mm.

Figure 6 Scapholeberis kingii Sars, 1888, parthenogenetic females from Farm Dam, New South Wales, Australia.

(A) Thoracic limb I. (B) Thoracic limb II. (C) Thoracic limb III. (D) Thoracic limb IV. (E) Thoracic limb V. Scale bar = 0.1 mm.

Figure 7 Scapholeberis kingii Sars, 1888, parthenogenetic and ephippial females from Farm Dam, New South Wales, Australia.

(A–E) Parthenogenetic females, (F–L), Ephippial females. (A) Ephippial female, lateral view. (B) Valve, inner view. (C) Posteroventral portion of valve, inner view. (D) Head, ventral view. (E) Antenna I. (F) I, Ephippial females, lateral view. (G) J, Ephippia, lateral view. K, Head, lateral view. (H) L, Ornamentation of central portion of ephippia. Scale bars = 0.2 mm for (A), (B), (F), (G), (I) and (J), 0.1 mm for (D) and (K), 0.05 mm for (H), 0.02 mm for (C), (E) and (L).

Figure 8 Scapholeberis kingii Sars, 1888, ephippial females from Farm Dam, New South Wales, Australia.

(A) Ephippial female, dorsal view. (B) Ephippium, dorsal view. (C) Ephippium, dorsal view on higher magnification. (D) Head, dorsal view. (E) Ephippial female, ventral view. (F) Head, ventral view. (G) Head on higher magnification, ventral view. Scale bars = 0.2 mm for (A), (B) and (E), 0.1 mm for (C) and (D), 0.05 mm for (F) and (G).

Figure 9 Scapholeberis kingii Sars, 1888, preephippial female from the roadside pool near Lake Bantic, West Coast, Tasmania, Australia.

(A) Preephippial female, lateral view. (B) Head, lateral view. (C) Postabdominal claw, lateral view. (D) Posterior portion of body. (E) and (F), Posterior portion of body on higher magnifications. Scale bars = 0.2 mm for (A) and (D), 0.1 mm for (B) and (E), 0.05 mm for (F) and 0.02 mm for (C).

Daphnia mucronata (Müller) in King, 1853, p. 255–265, fig. 6E.

Scapholeberis kingii Sars, 1888, p. 68.

Scapholeberis kingi Sars in Henry, 1919, p. 465; Henry, 1922, p. 29, Pl. 4: Fig. 3; Dumont, 1983, 105–106, Pl. 3; Dumont & Pensaert, 1983, p. 24–25, Fig. 2: 3; Fig. 4: 4; Fig. VI: 1–2; Pl. 1: 8; Pl. 2: 4; Pl. 3: 5, 7, 9; Pl. 4: 1–7; Pl. 5: 1–2, 4; Fig. 10: 3; Pl. 6: 6–8; Fig. 12 Fig. 21: 4 (partial); Smirnov, 1995, p. 5; Shiel & Dickson, 1995, p. 35.

Figure 10 Scapholeberis intermedius Daday, 1898, parthenogenetic females from Collectio Dadayana.

(A) Adult parthenogenetic female, lateral view (DAD 10-70-159). (B) Juvenile parthenogenetic female, lateral view (DAD 10-70-156). (C) Head, dorsal (?) view (DAD 10-70-156). (D) Antenna II (DAD 10-70-156). Scale bars = 0.1 mm.

? Scapholeberis Kingi n. sp. in Sars, 1903, p. 8–10, Pl. 1: Fig. 2a–c. – junior homonym of S. kingi Sars, 1888.

Type locality. “South Creek” and “Paramatta” (King, 1853), New South Wales, Australia.

Type material. Lost.

Material studied here. See Table S2.

Redescription. Parthenogenetic female (Figs. 4A–4E, 5, 6 and 7A–7E). In lateral view body relatively elongated, dorsal margin regularly arched, ventral margin almost straight, maximum height at body midpoint (body height/length ratio about 0.6 for adults and 0.5 for juveniles) (Figs. 4A and 7A). In dorsal or ventral view body ovoid, moderately compressed from sides (Fig. 4B). In anterior view body moderately compressed, dorsal keel absent. Posterodorsal angle obtuse, posteroventral angle almost straight, with a long spine (mucro) (Figs. 4A, 5D, 5E and 7A–7C). A row of numerous small setules on inner face of posterior margin of valve (Figs. 5D, 5E, 7B and 7C). Ventral margin covered by setae of different size (Figs. 5A–5D). Anterovenral angle of valve broadly rounded, its ventral portion with a small protuberance (Fig. 7B). Valves with well-developed sculpture of polygonal reticulation.

Head large for a daphniid (Figs. 4A and 7A). In lateral view head elongated, with a prominent rostrum, its distal portion roundish (Figs. 4A and 7A). In dorsal view head elongated, head shield with low lateral projections (fornices) covering bases of antennae II, a sclerotized ridge departs from the insertion of antenna II and extends to the side of head (Fig. 4B). In anterior view head slightly compressed from lateral sides (Figs. 4C and 7D). In ventral view postero-ventral portion of head forms a three-lobed rostrum, due to a shallow depression in points of antenna I insertion on each side, its middle lobe rounded, with a minute frontal head pore (Figs. 4C, 7D and 7E). In anterior view, distance between the center of ocellus and eye slightly greater (almost twice) than distance from the center of ocellus to the tip of rostrum (Fig. 4C). Dorsal head pores absent. Labrum large, distal labral plate with bunches of long setules, in ventral view labrum triangular, with lateral projections (Figs. 4D and 7D).

Valve with straight ventral margin (Figs. 4A, 5D, 7A and 7B). Adhesive ventral rim of valves modified into “sucker-plate” (in terms of Dumont & Pensaert, 1983), no setae along most part of the sucker length except few rarely located setae at anteriormost portion and several sparsely located setae at posterior portion near mucro (Figs. 5A–5C). Inner surface of posterior margin with a broad “hyaline membrane” (in terms of Dumont & Pensaert (1983)) extending the posterior rim and a “denticulated membrane” (in terms of Dumont & Pensaert (1983)) consisting of row of short setules along the posterior rim (Figs. 5D, 5E, 7B and 7C).

Thorax relatively long for daphniids, abdomen short (Fig. 4A).

Postabdomen almost rectangular, postabdomen length/height ratio about three (Figs. 5F and 5G). Ventral margin almost straight. Preanal margin two times longer than anal margin. Anal and postanal margins almost equal in length. Basis of claws slightly inflated, bordered from distal margin by a clear incision (Figs. 5G–5I). Postanal portion of postabdomen armed with long, thin solitary teeth and bunches of fine setules (Figs. 5G and 5H). Bunches of fine setules also on anal margin and lateral surface of postabdomen. Postabdominal claw long (almost as long as anal margin), slightly curved (Figs. 5H and 5I). Its external side armed by three rows of small denticles, decreasing in size distally. Denticles in middle portion of claw are stronger and located more sparsely as compared to other denticles. Basal spine absent (Figs. 5H and 5I).

Antenna I jointed to the head surface, relatively short, antennular body with aesthetascs exceeds tip of rostrum in length (Figs. 5J, 5K, 7D and 7E). Antennular sensory seta slender, arising subdistally, almost equal in length to antennular body. Nine aesthetascs unequal in size (Figs. 5J, 5K and 7E). All aesthetasc tips projecting beyond tip of rostrum.

Antenna II relatively long (Figs. 4A, 5L, 5M and 7A). Antennal formula for setae: 0-0-1-3/1-1-3. Antennal formula for spines: 0-1-0-1/0-0-1. Coxal part folded, with two sensory setae. Basal segment elongated, covered by concentric rows of fine setules with a very thin spine between antenna II exopod and endopod branches on outer surface and a short bisegmented seta on outer surface (Figs. 5L and 5M). Branches relatively elongated, all segments cylindrical, covered by concentric rows of fine setules and tiny denticles around their distal margins. Apical setae typical for daphniids (as long as antennal branches), setulated asymmetrically. Lateral setae arising from basal and middle endopod segment long (reach tips of apical setae) (Fig. 5L). Lateral seta arising from third exopod segment thin and relatively short (reaches the middle of apical setae). Spine on the second exopod segment short and thin. Spines on apical segments of endopod and exopod branches very small and short, subequal in size to concentric apical denticles, arising from distal portions of apical segments.

Thoracic limbs: five pairs (Figs. 6A–6E).

Limb I with ovoid epipodite (Fig. 6A). Accessory setae long, armed by long setules. Outer distal lobe with two setae unequal in size. Distal segment of the longest seta unilaterally armed by short setules; proximal portion of this seta bears especially long setules. Shorter seta of outer distal lobe bilaterally armed by short setules. Inner distal lobe (endite 5) with three setae unequal in size and shape (Fig. 5A: 1, 1′, 1″). Two setae bisegmented, with elongated distal portions. A single seta 1 brush-shaped (in terms of Dumont & Pensaert (1983)), its distal end abrupt, bearing long thickened setules. Endite 4 with a short anterior seta 2 and two posterior setae (Fig. 6A: a–b). The ratio between seta 1′ and seta 2 is almost 2.5 (i.e., seta 2 is relatively short as compared to S. cf. intermedius from Africa, see below). Endite 3 with a short and thin anterior seta 3 and two posterior setae (Fig. 6A: c–d). Endite 2 with a short anterior seta 4 and four posterior setae (Fig. 6A: e–h). Two ejector hooks subequal in size.

Limb II large (Fig. 6B). Limb distal portion (exopodite) as large ovoid setulated lobe with two soft setae unequal in size. Four endites fused (e5–e2), bearing in toto six setae. Distal segments of anterior setae a–d covered by short denticles. Two posterior setae (Fig. 6B: a, d) bear long setules. Gnathobase (endite 5) with two rows of setae: four anterior setae (Fig. 6B: 1–4, among them seta 1 as a small elongated sensillum) and six posterior setae of gnathobasic “filter plate”.

Limb III with a large ovoid epipodite (Fig. 6C) and a flat round exopodite bearing four distal setae (Fig. 6C: 1–4), (among them seta 2 the longest) and two lateral setae (Fig. 6C: 5–6) unequal in length. Setae 3–5 covered by long setules. Setae 1–2 featured by long setules in their proximal portions and bearing shorter stiff setules on their distal segments. Inner distal portion of limb with four endites: endite 5 with a single, short anterior seta (1) and a posterior seta (a); endite 4 with a single anterior seta (2) and a single posterior (b) seta; endite 3 with a short anterior seta (3) and two posterior setae (c–d); endite 2 with two anterior seta (4–5?) and four posterior (e–h) setae. The rest of limb inner-distal portion as a singular large lobe, modified gnathobase, bearing numerous posterior soft setae, each with chitinous insertion within basal portion of distal segment, and a single, relatively long anterior seta (1) in its distal corner (Fig. 6C).

Limb IV with a large ovoid epipodite (Fig. 6D) and wide, flat rounded exopodite with two protruding setulated lobes, four distal (Fig. 6D: 1–4) and two lateral (Fig. 6D: 5–6) setae. Among them seta 4 the longest. Inner-distal portion of this limb with completely fused endites, distally with two setae (Fig. 6D: 1–2) of unclear homology, the most part of limb inner margin is a gnathobase filter plate consisting of numerous posterior setae.

Limb V (Fig. 6E) with a setulated preepipodite, large, subovoid epipodite, triangular exopodite supplied with two small, thin distal setae (Fig. 6E: 1–2) unequal in length and a large lateral seta (Fig. 6E: 3). Inner limb portion as an ovoid flat lobe, with setulated inner margin and a single, large seta.

Ephippial female (Figs. 4F–4H, 7F–7L and 8A–8G). Body shape in general as in parthenogenetic female. Dorsal portion of valves modified into ephippium. Ephippium dark brown, ovoid, clearly bordered from ventral and lateral portions of valves separating during its casting off (Figs. 4F, 7F–7G, 7I and 7J). Egg chamber with a single egg, elongated, its sculpture represented by shallow depressions (Figs. 4F, 4G, 7H, 7L and 8C). Sculpture of the rest of ephippium is represented by small polygons. Lateral keels are well distinguishable from the lateral (Figs. 4F, 4G, 7F, 7G, 7I and 7J) and dorsal view (Figs. 8A and 8B). From the dorsal view, area between two keels strongly elongated, keels not projected laterally out of body dorsal contour (Figs. 8A and 8B).

Preephippial female (Figs. 9A–9F). Body shape in general similar to that in parthenogenetic female (Fig. 7A). Lateral keels already visible (Figs. 9A, 9D and 9E), but dorsal portion of valves almost weekly chitinized. Ventral and lateral borders between preephippium and the rest of valves not developed (Figs. 9A and 9D).

Male. Despite significant sampling effort, we failed to detect males in the investigated samples. Although males of Scapholeberis have been described by Dumont & Pensaert (1983), it is difficult to detect them in nature or in laboratory cultures. In general view, males are similar to juvenile females and could not be distinguished without dissection. Also, it seems possible, that at least in some Scapholeberis species, ephippial females may appear in the natural populations and under laboratory conditions without males. The same situation is known for some Daphnia O.F. Mueller, 1785 (Kotov, 2013).

Size. Medium-sized species, parthenogenetic female up to 0.55 mm in length without mucro (and 0.57 mm with mucro), ephippial female up to 0.57 mm in length without mucro (and 0.61 with mucro).

Variability. No significant variability was found among the investigated individuals.

Taxonomic notes. King (1853, p. 255-256, plate V, fig. e) found “Daphnia mucronata (Müller)” in “South Creek” and “Paramatta”, New South Wales, Australia. In his diagnosis, he mainly reproduced the previous redescription of Scapholeberis mucronata by Baird (1850, p. 99–100) made for European populations, but pointed on two differences of the Australian specimens: (1) “European specimens have the upper part of the head sometimes terminated by a sharp-curved point, and directed upwards. I have not found any such variety here”; (2) “the head of each of Baird’s figures is larger than that of the Australian species”. Sars (1888: p. 68) took these differences into his consideration and established new taxon, S. kingii Sars, 1888, referring to the description of King (1853) rather than based on his own original material. It is an acceptable action according to the International Commission on Zoological Nomenclature (ICZN) (2000). Specimens of this taxon from Australia are absent from the collection of G.O. Sars in the Zoological Museum of the Oslo University, Norway. King’s specimens were eligible to be designated as types for S. kingii International Commission on Zoological Nomenclature (ICZN) (2000), but the specimens were apparently lost.

Then Sars (1903, p. 8–10, plate 1, figs 2, 2a, 2b) proposed the name “Scapholeberis Kingi”, G.O. Sars, n. sp. “for populations from Sumatra (unknown water bodies in “territories of Deli and Langkat” collected by Mr. Iversen) with the following explanation: “The above-described species is unquestionably identical with the Australian form recorded by King as Daphnia mucronata. It is certainly very nearly allied to the European species, but apparently specifically distinct, differing, as it does, not only in the much smaller size, but also in the shape of the head and in the less sharply angulated anterior part of the valves. The sculpture of the shell is, moreover, much coarser than in the European species”. But, Sars’ earlier species name “S. kingii” of Australia has precedence over the Sumatran species. The Sumatran specimens are present in the Collection of G.O. Sars (GOS F 9540, GOS F 12272, GOS F 12880). However, these specimens are not regarded as types because they were not reported in the original taxon description. According to the drawings of Sars (plate 1, figs 2, 2a, 2b), the specimens from Sumatra belong to the S. kingii group. Presently it is unknown if the populations from Sumatra belong to S. kingii s.str., S. smirnovi sp.nov., or another taxon (tropical Asian populations are not revised here).

Dumont & Pensaert (1983) correctly pointed out that Dumont (1983) erroneously stated that S. kingi Sars, 1888 was a nomen nudum (and claimed that the species should have been named S. kingi Sars, 1903).

Distribution. To date, we can confirm its presence in Australia only, where it is a common taxon (Dumont, 1983; Smirnov, 1995; Shiel & Dickson, 1995), but we cannot fully exclude the chance that there are several additional taxa within this group.

Records of S. kingii from Spain, Sicily and Central Europe have been declared dubious (Alonso, 1996; Marrone, Barone & Naselli-Flores, 2005; Hudec, 2010), but members of the S. kingii species group (see below) were found to be common in Northern Africa (Ghaouaci et al., 2018; Neretina, 2018). In the Eastern Palearctic, the range of S. cf. kingii extends northwards, up to Japan (Tanaka, 1998a, 1998b), the Korean Peninsula (Kotov, Jeong & Lee, 2012) and the Russian side of the Amur River (=Heilong Jiang in Chinese) basin (Kotov et al., 2011). Therefore, the S. kingii species complex is regarded as a typical “tropicopolitan” taxon with a very wide geographic range in the Eastern Hemisphere.

2. Scapholeberis intermedius Daday, 1898

Figure 10

Scapholeberis mucronata var. intermedia Daday, 1898, p. 59–60, Fig. 29a–b.

? Scapholeberis kingi Sars in Gurney, 1907, p. 277–278; Fernando, 1980, p. 97; Michael & Sharma, 1988, p. 73–74, Fig. 20a–c; Chatterjee et al., 2013, p. 20–21.

Type locality. “Sümpfe der Umgebung des Kalawewa-Sees”, Sri Lanka (Daday, 1898).

Type material (studied here). See Table S2.

Brief redescription of museum material. Redescription. Parthenogenetic female. In lateral view body elongated and ovoid, dorsal margin regularly arched, ventral margin straight, maximum height at middle of body (body height/length ratio about 0.61 for adults and 0.59 for juveniles) (Figs. 10A and 10B). Head large with well developed rostrum (Figs. 10A and 10B). Posterodorsal angle obtuse, posteroventral angle almost straight with long mucro (Figs. 10A and 10B). Posterior margin generally almost straight or slightly curved. Ventral margin almost straight. Anterovenral angle broadly rounded, its ventral side with small protuberance.

Head large (Figs. 10A and 10B). In lateral view head elongated with prominent rostrum. Distal portion of rostrum roundish. Compound eye large, ocellus is not recognizable (Figs. 10A and 10C).

Antenna II relatively long, endopod branch slightly longer than exopod (Fig. 10D). Antennal formula identical to previous species.

Ephippial female, male. Completely absent in the type material.

Size. Medium-sized species, parthenogenetic female up to 0.62 mm in length without mucro (and 0.63 mm with mucro).

Variability. No significant variability was found in the investigated individuals.

Taxonomic remarks. According to Daday (1898) this “variety” has intermediate morphological characters between S. mucronata O.F. Müller and S. obtusa Schödler. The latter is now regarded as a junior synonym of Megafenestra aurita Fischer. Unfortunately, type material of S. intermedius is represented by permanent slides with parthenogenetic females in the lateral or almost lateral position (Fig. 10). Gamogenetic females and males are completely absent in the type series. Thus, we have no opportunity to compare the morphological features (proportions of head and shape of ephippium from the dorsal position) of typical S. intermedius, S. smirnovi sp.nov. and African S. cf. intermedius (see below). Based on the genetic data, we demonstrated that populations from Ethiopia and the Russian Far East form unique lineages (Figs. 1 and 2). We propose here that S. smirnovi sp.nov. is a separate taxon, well delineated from other S. kingii-like species (see below). Morphological and genetic investigations of kingii-like populations from the type locality of S. intermedius, Sri Lanka (and South Asia as a whole) will be carried out in future studies. To date we have no suitable material of S. kingii with ephippial females from this area.

3. Scapholeberis cf. intermedius Daday, 1898

Figures 11–15

Figure 11 Scapholeberis cf. intermedius Daday, 1898, a parthenogenetic female from Bahir Dar Bay of Lake Tana, Amhara, Ethiopia.

(A) Parthenogenetic female, lateral view. (B) Head, lateral view. (C) Head, ventral view. (D) Labrum. (E) Valve. (F)–(H) Armature of posteroventral angle of valve. (I) Postabdomen. (J) Distal portion of postabdomen. (K) Postabdominal seta. (L) Antenna I. Scale bars = 0.1 mm.

Figure 12 Scapholeberis cf. intermedius Daday, 1898, a parthenogenetic female from Bahir Dar Bay of Lake Tana, Amhara, Ethiopia.

(A) Distal portion of postabdomen. (B) Antenna II. (D–J) Fragments of antenna II. (K) Thoracic limb I. Scale bars = 0.1 mm.

Figure 13 Scapholeberis cf. intermedius Daday, 1898, a parthenogenetic female from Bahir Dar Bay of Lake Tana, Amhara, Ethiopia.

(A) Thoracic limb I. (B) Thoracic limb II. (C) and (D) Fragments of thoracic limb II. (E) Thoracic limb III. (F) and (G) Fragments of thoracic limb III. Scale bars = 0.1 mm.

Figure 14 Scapholeberis cf. intermedius Daday, 1898, a parthenogenetic female from Bahir Dar Bay of Lake Tana, Amhara, Ethiopia.

(A) Thoracic limb IV. (B) and (C) Fragments of thoracic limb IV. (D) Thoracic limb V. (E) Fragment of thoracic limb V. Scale bars = 0.1 mm.

Figure 15 Scapholeberis cf. intermedius Daday, 1898, a parthenogenetic female from Bahir Dar Bay of Lake Tana, Amhara, Ethiopia.

(A) Parthenogenetic female, lateral view. (B) Anterior portion of body. (C) Head, lateral view. (D) Posterior portion of body. (E), Ornamentation of valve. Scale bars 0.2 mm for (A) and (D), 0.1 mm for (B), 0.05 mm for (C) and (E).

? Scapholeberis kingi Sars in Sars, 1916, p. 314–315, Pl. XXXII: 3, 3a, 3f; Brehm, 1938, p. 489; Gauthier, 1951, p. 48–50, text-figure in p. 49, C–D; Harding, 1961, p. 40; Rey & Saint-Jean, 1969, p. 26, Fig. 5a–c; Dumont & Van de Velde, 1977, p. 80; Dumont, Laureys & Pensaert, 1979, p. 265, 267; Day et al., 1999, p. 97, Fig. 4.6.B.

Material studied here. See Table S2.

Description. Parthenogenetic female (Figs. 11–15). In lateral view, body regularly elongated, dorsal margin broadly arched, ventral margin almost straight, maximum height at middle of body (body height/length ratio about 0.59 for adults, juveniles not studied) (Figs. 11A and 15A). In dorsal and ventral view body ovoid, only moderately compressed from sides. In anterior view body moderately compressed, dorsal keel absent. Head large with well developed rostrum (Figs. 11A, 11B and 15A–15C). Depression between head and rest of body absent, but dorsal contour may be slightly concave under compound eye and antenna. Posterodorsal and posteroventral angles expressed (Figs. 11A, 11E, 15A and 15D). Posterodorsal angle obtuse, posteroventral angle almost straight with long mucro (Figs. 11A, 11E, 15A and 15D). Posterior margin generally almost straight or slightly curved. A raw of numerous small setules on inner face of posterior margin of valve (Figs. 11F and 11G). Ventral margin almost straight, covered by setae of different size (Fig. 11E). Anteroventral angle broadly rounded, its ventral side with small protuberance (Figs. 11A, 11E, 15A and 15D). Valves with developed sculpture, consisting of polygons (Figs. 11E, 15D and 15E).

Head large for daphniids (Figs. 11A, 11B and 15A–15C). In lateral view head elongated, with a prominent rostrum. Distal portion of rostrum roundish. In anterior view, head elongated and round, slightly compressed from lateral sides (Fig. 11C). Its ventral portion three-lobed with depression for antennulae. A central lobe is rostrum, its tip broadly rounded with small shallow incision. In anterior view, distance between the center of ocellus and eye significantly greater (almost in three times) than distance from the center of ocellus to the tip of rostrum (Fig. 11C). Dorsal head pores absent, frontal head pore was not studied. Labrum large (Fig. 11D). Distal labral plate with bunches of long setules.

Valve with straight ventral margin (Figs. 11E). Adhesive ventral rim of valves modified into “sucker-plate”. Inner surface of posterior margin with a broad “hyaline membrane” (in terms of Dumont & Pensaert, 1983) extending the posterior rim and a “denticulated membrane” (in terms of Dumont & Pensaert, 1983) consisting of row of short setules along the posterior rim (Figs. 11F and 11G).

Postabdomen almost rectangular, slightly narrowing distally; postabdomen length/height ratio about 2.6 (Fig. 11I). Ventral margin straight. Preanal margin three times longer than anal margin. Anal and postanal margins almost equal in length. Base of claws not inflated (Figs. 11I, 11J and 12A). Postanal portion of postabdomen armed with long and thin denticles and bunches of fine setules. Bunches of fine setules also on anal margin and lateral surface of postabdomen. Postabdominal claw long (almost as long as anal margin), slightly curved (Figs. 11I, 11J and 12A). Its external side armed by three rows of small denticles, deceasing in size distally. Basal spine absent (Figs. 11I, 11J and 12A).

Antenna I relatively short, antennular body with aesthetascs exceeds tip of rostrum in length (Fig. 11L). Nine aesthetascs unequal in size.

Antenna II relatively long (Figs. 11A and 12B–12J). Antennal formula for setae: 0-0-1-3/1-1-3. Antennal formula for spines: 0-1-0-1/0-0-1. General structure of antenna II identical to species described above.

Thoracic limbs: five pairs.

Limb I (Figs. 12K and 13A). Accessory setae very long, prominent. Outer distal lobe with two setae unequal in size. Distal segment of the longest seta unilaterally armed with short setules; proximal portion of this seta bears especially long setules. Shorter seta of outer distal lobe bilaterally covered by short setules. Inner distal lobe (endite 5) with three setae unequal in size and shape (Figs. 12K and 13A: 1, 1′, 1″). Endite 4 with a short anterior seta 2 and two posterior setae (Figs. 12K and 13A: a–b). The ratio between seta 1′ and seta 2 is almost 1.5 (i.e., seta 2 is relatively long in the comparison of other Scapholeberis species investigated here, see redescription of S. kingii above and description of S. smirnovi sp.nov. below). Endite 3 with a short and thin anterior seta 3 and two posterior setae (Figs. 12K and 13A: c–d). Endite 2 with a short anterior seta 4 and four posterior setae (Figs. 12K and 13A: e–h). Two ejector hooks almost similar in size.

Limb II large, basically similar to other Scapholeberis species investigated here (Figs. 13B–13D).

Limb III (Figs. 13E–13G) with a large ovoid epipodite and a flat round exopodite bearing four distal setae, (among them seta 2 the longest, Figs. 13E and 13F) and two lateral setae unequal in length. Setae 3–5 covered by long setules. Setae 1–2 armed with long setules in their proximal portions and bear shorter stiff setules on their distal segments. Inner distal portion of limb (Figs. 13E and 13G) with four endites: endite 5 with a single, short anterior seta (1) and a posterior seta (a); endite 4 with a single anterior seta (2) and a single posterior (b) seta; endite 3 with a short anterior seta (3) and two posterior setae (c–d); endite 2 with two anterior seta (4–5) and four posterior (e–h) setae. The rest of limb inner-distal portion as a singular large lobe, modified gnathobase, bearing numerous posterior soft setae, each with chitinous insertion within basal portion of distal segment, and a single, relatively long anterior seta (1) in its distal corner. Also, two small sensillae recognizable in this portion.

Limb IV (Figs. 14A–14C) with a large ovoid epipodite and wide, flat rounded exopodite with two protruding setulated lobes, four distal and two lateral setae. Among them seta 4 the longest (Figs. 14A and 14B). Inner-distal portion of this limb with completely fused endites, distally with two setae of unclear homology, the most part of limb inner margin is a gnathobase filter plate consisting of numerous posterior setae (Fig. 14C). Also, two small sensillae recognizable in this portion.

Limb V (Figs. 14D and 14E) with a setulated preepipodite, large, subovoid epipodite, triangular exopodite supplied with two small, thin distal setae (Figs. 14D and 14E: 1–2) unequal in length and a large lateral seta (Figs. 14D and 14E: 3). Inner limb portion as an ovoid flat lobe, with setulated inner margin and a single, large seta. A small sensillum recognizable near seta 2.

Ephippial female, male. Despite significant efforts, we did not find gamogenetic females and males in African localities. Other authors who dealt with the description of African populations also did not observe Scapholeberis ephippial females and males in their materials.

Size. Medium-sized species, parthenogenetic female up to 0.70 mm in length without mucro (and 0.73 mm with mucro).

Variability. No significant variability was found among the investigated individuals.

Other records in Africa. Distribution of Scapholeberis in Africa remains scarcely studied. Reliable records of S. kingii populations are known from West Africa (Dumont, 1981; Egborge, Onwudinjo & Chigbu, 1994; Chiambeng & Dumont, 2005), Central Africa (Rey & Saint-Jean, 1969), and South Africa (Sars, 1916; Day et al., 1999).

4. Scapholeberis smirnovi sp. nov.

Figures 16–20

Figure 16 Scapholeberis smirnovi sp.nov. from the puddle near Lake Maloe Utinoe, Primorski Territory, Far East, Russia.

(A–D) Adult parthenogenetic females, (E and F) Juvenile parthenogenetic female, (G–I) Ephippial females. (A) Adult parthenogenetic female, lateral view. (B) Parthenogenetic female, dorsal view. (C), Head, ventral view. (D) Labrum. (E) Juvenile parthenogenetic female. (F) Head, ventral view. (G) Ephippial female, lateral view. (H) Ephippial female, dorsal view. (I) Ornamentation of ephippium. Scale bars = 0.1 mm.

Figure 17 Scapholeberis smirnovi sp.nov. from the puddle near Lake Maloe Utinoe, Primorski Territory, Far East, Russia.

(A), Valve, outer view. (B) Valve, ventral view. (C and D) Armature of valve. (E) Posteroventral portion of valve, inner view. (F–H), Postabdomen. (I), Postabdominal claw. (J and K), Antenna I. (L and M), Antenna II. Scale bars 0.1 = mm.

Figure 18 Scapholeberis smirnovi sp.nov. from the puddle near Lake Maloe Utinoe, Primorski Territory, Far East, Russia.

(A and B) Thoracic limb I. (C and D) Thoracic limb II. (E) Thoracic limb III. (F and G) Thoracic limb IV. (H) Thoracic limb V. Scale bar = 0.1 mm.

Figure 19 Scapholeberis smirnovi sp.nov., ephippial and preephippial females from the puddle near Lake Maloe Utinoe, Primorski Territory, Far East, Russia.

(A, B, D and F) Ephippial females, (C) Preephippial female. (A) Ephippial female, lateral view. (B) Ephippium, lateral view. (C) Preephippial female, lateral view. (D) Ephippial female, anterodorsal view. (E) Ephippium, anterodorsal view. (F) Ephippial female, ventral view. Scale bars = 0.2 mm for (A–D, F), 0.1 mm for (E).

Figure 20 Scapholeberis smirnovi sp.nov., ephippial females from a pond in Choenggye Mountains, Seoul, the Republic of South Korea.

(A) Ephippial female, lateral view. (B and C) Head, lateral view. (D) Ephippium, lateral view. (E and F) Ornamentation of ephippium. (G) Ephippial female, dorsal view. (H) Head, dorsal view. (I and J) Ephippium, dorsal view. (K and L) Armature of ephippium on higher magnifications. Scale bars = 0.2 mm for (A, D, G and I), 0.1 mm for (B, C, H, and J–L), 0.5 mm for (F), 0.2 mm for (E).

Figure 21 Analysis of identification for four species groups of Scapholeberis based on GenBank data.

Scapholeberis kingi Sars in Uéno, 1940, p. 342; Tanaka, 1998a, p. 30–31, Figs. 2A–2C; Tanaka, 1998b: p. 15–16, Fig. 9–10; Tanaka, Ohtaka & Nishino, 2004, p. 173–174, Fig. 3; Kotov et al., 2011, p. 403, Table 1; Kotov, Jeong & Lee, 2012, p. 58, Fig. 5; Jeong, Kotov & Lee, 2014, p. 219.

? (at least partially) Scapholeberis kingi Sars in Chiang & Du, 1973, p. 145–146, Fig. 97a-c; in Du Nan-shan, 1973, p. 44, Fig. 13; Xiang et al., 2015, p. 13–14.

Scapholebeis mucronata (O.F. Müller) in Uéno, 1927, p. 281, Fig. 9 (not 9a–9e!);

Scapholeberis rammneri Dumont & Pensaert in Yoon, 2010, p. 64–66, Fig. 34.

Publication Zoobank ID. See nomenclatural acta above.

Zoobank taxon ID. urn:lsid:zoobank.org:act:62ABBAFB-249D-453A-BB8D-E59ECB1AB2B0.

Etymology. The taxon is named after Professor Nikolai N. Smirnov, a renowned Russian zoologist and hydrobiologist, who established the Russian school of cladocerology and made large advances in the study of freshwater zooplankton.

Type locality. A puddle near Lake Maloe Utinoe (N 43.4127°, E 131.8214°), Primorski Territory, the Russian Far East.

Type material. Holotype: an ephippial female, fixed in 96% ethanol, deposited at the collection of Zoological Museum of Moscow State University, MGU Ml-189. The label of holotype is: “Scapholeberis smirnovi sp. nov., 1 ephippial female from puddle near Lake Maloe Utinoe, Holotype”. Paratypes. See Table S2.

Description. Parthenogenetic female (Figs. 16A–16F). In lateral view body relatively elongated, dorsal margin regularly arched, ventral margin almost straight, maximum height at body middle (body height/length ratio about 0.6 for adults and 0.5 for juveniles) (Figs. 16A and 16E, correspondingly). In dorsal or ventral view body ovoid, moderately compressed from sides (Fig. 16B). In anterior view body moderately compressed, dorsal keel absent. Posterodorsal angle obtuse, posteroventral angle almost straight, with a long spine (mucro) (Figs. 16A, 16E and 17A–17E). A row of numerous small setules on inner face of posterior margin of valve (Fig. 17E). Ventral margin covered by setae of different size (Figs. 17A–17D). Anteroventral angle of valve broadly rounded, its ventral portion with a small protuberance (Figs. 16A and 16E). Valves with well-developed sculpture of polygonal reticulation.

Head large for a daphniid (Fig. 16A). In lateral view head elongated, with a prominent rostrum, its distal portion roundish (Fig. 16A). In dorsal view head elongated, head shield with low lateral projections (fornices) covering bases of antennae II, a sclerotized ridge departs from the insertion of antenna II and extends to the side of head. In anterior view head slightly compressed from lateral sides. In ventral view postero-ventral portion of head forms a three-lobed rostrum, as there is a shallow depression at insertion points of antenna I on each side, its middle lobe rounded, with a minute frontal head pore (Figs. 16C). In anterior view, distance between the center of ocellus and eye significantly greater (almost in five times) than distance from the center of ocellus to the tip of rostrum (Figs. 16F). Dorsal head pores absent. Labrum large (Fig. 16D), similar to other Scapholeberis species.

Valve with straight ventral margin (Figs. 16A and 17A). Adhesive ventral rim of valves modified into “sucker-plate” (Figs. 16A–16D), details of its structure identical to S. kingii.

Thorax relatively long, abdomen short (Fig. 16A).

Postabdomen almost rectangular, postabdomen length/height ratio about 2.8 (Figs. 17F–17H). Ventral margin almost straight. Preanal margin two times longer than anal margin. Anal and postanal margins almost equal in length. Base of claws slightly inflated, bordered from distal margin by a clear incision (Figs. 17G–17I). Postanal portion of postabdomen armed with long, thin solitary teeth and bunches of fine setules. Bunches of fine setules also on anal margin and lateral surface of postabdomen. Postabdominal claw long (almost as long as anal margin), slightly curved (Figs. 17G–17I). Its external side armed by three rows of small denticles, decreasing in size distally. Denticles in middle portion of claw are stronger and distributed more sparsely as compared to other denticles. Basal spine absent (Figs. 17G–17I).

Antenna I relatively short, its proportions similar to other Scapholeberis species (Figs. 17J and 17K). Nine aesthetascs unequal in size.

Antenna II relatively long (Figs. 16A, 17L and 17M). Antennal formula for setae: 0-0-1-3/1-1-3. Antennal formula for spines: 0-1-0-1/0-0-1. Fine armature of antenna II similar to S. kingii.

Thoracic limbs: five pairs (Figs. 18A–18H).

Limb I with ovoid epipodite (Figs. 18A and 18B). Accessory setae long, armed by long setules. Outer distal lobe with two setae unequal in size. Distal segment of the longest seta unilaterally armed by short setules; proximal portion of this seta bears especially long setules. Shorter seta of outer distal lobe bilaterally armed by short setules. Inner distal lobe (endite 5) with three setae unequal in size and shape (Fig. 18A: 1, 1′, 1″). Two setae bisegmented, with elongated distal portions. A single seta 1 brush-shaped (in terms of Dumont & Pensaert (1983)), its distal end abrupt, bearing long thickened setules. Endite 4 with a short anterior seta 2 and two posterior setae (Fig. 18A: a–b). The ratio between seta 1′ and seta 2 is almost 2.5 (i.e., seta 2 is relatively short as compared to S. cf. intermedius from Africa, and comparable to S. kingii, see above). Endite 3 with a short and thin anterior seta 3 and two posterior setae (Fig. 18A: c–d). Endite 2 with a short anterior seta 4 and four posterior setae (Fig. 18A: e–h). Two ejector hooks subequal in size.

Limb II large (Figs. 18C and 18D). Limb distal portion (exopodite) as large ovoid setulated lobe with two soft setae unequal in size. Four fused endites (e5–e2) bear six setae. Distal segments of anterior setae a–d covered by short denticles. Two posterior setae (a and d) bear long setules. Gnathobase (endite 5) with two rows of setae: four anterior setae (Fig. 18C: 1–4, among them seta 1 as a small elongated sensillum) and six posterior setae of gnathobasic “filter plate”.

Limb III with a large ovoid epipodite (Fig. 18E) and a flat round exopodite bearing four distal setae (Fig. 18E: 1–4), (among them seta 2 the longest) and two lateral setae (Fig. 18E: 5–6) unequal in length. Proportions and armature of all setae similar to S. kingii.

Limb IV with a large ovoid epipodite (Figs. 18F and 18G) and wide, flat rounded exopodite with two protruding setulated lobes, four distal (Fig. 18F: 1–4) and two lateral (Fig. 18F: 5–6) setae. Proportions and armature of all setae similar to S. kingii.

Limb V (Fig. 18H) with a subovoid epipodite, triangular exopodite supplied with two small, thin distal setae (Fig. 18H: 1–2) unequal in length and a large lateral seta (Fig. 18H: 3). Inner limb portion as an ovoid flat lobe, with setulated inner margin and a single, large seta.

Ephippial female (Figs. 16G–16I, 19A, 19B, 19D–19F and 20A–20L). Body shape in general as in parthenogenetic female. Dorsal portion of valves modified into ephippium. Ephippium dark brown, ovoid, clearly bordered from ventral and lateral portions of valves separating during its casting off (Figs. 16G, 19A, 19B, 20A and 20D). Egg chamber with a single egg, elongated, its sculpture represented by shallow depressions (Figs. 16G and 20F). Sculpture of the rest of ephippium is represented by small polygons. Lateral keels are well distinguishable from the lateral (Figs. 16G, 19A, 19B and 20A–20D) and dorsal view (Figs. 16H, 19E, 20G and 20I–20L). From the dorsal view, area between two keels strongly rounded, keels strongly projected laterally out of body dorsal contour (Figs. 16H, 19D and 20G).

Preephippial female (Fig. 19C). Body shape in general similar to that in parthenogenetic female. Lateral keels already visible (Fig. 19C), but dorsal portion of valves weakly chitinized. Ventral and lateral borders between preephippium and the rest of valves not developed.

Male. Despite significant efforts, we did not find males in the investigated samples.

Taxonomic notes. Records of a “tropical” taxon, S. kingii, in northern regions such as South Korea and the Russian Far East surprised cladoceran investigators (Kotov, Jeong & Lee, 2012). However, we now know that the Far Eastern populations belong to a separate taxon, the real distribution of which needs to be accurately evaluated. To date, we had no DNA-available samples of S. cf. kingii from SE Asia, South China and Indian subcontinent where that taxon is common (Michael & Sharma, 1988; Korovchinsky, 2013; Kotov et al., 2013; Sinev, Gu & Han, 2015). Checking of the status of populations from different regions of the Palaeotropics is the next step in the revision of this group.

Our revision confirms again that the Far East of Eurasia, in its temperate portion, is an important source of new taxa, as it was already found previously (Kotov, Ishida & Taylor, 2009; Kotov et al., 2011).

Size. Medium-sized species, parthenogenetic female up to 0.75 mm in length without mucro (and 0.79 with mucro), ephippial female up to 0.70 mm in length (without mucro) (and 0.72 with mucro). Holotype 0.60 mm in length (without mucro), 0.37 mm in height.

Distribution. This taxon is known from the southern portion of the Far East of Russia, the Korean Peninsula, Japan and an adjacent region of China (Dongbei = Manchuria). It has also been recorded from a single locality in the southernmost portion of European Russia, but such a disjunct population may be due to an anthropogenic introduction.

Discussion

Comparison of the COI and 12S+16S phylogenies

The COI-based analyses reveal that the large genetic divergences within and among species groups of neustonic daphniids exist for both rRNA (Taylor, Connelly & Kotov, 2020) and protein-coding regions of the mitochondrial genome (the present study). Costa et al. (1997) reported a 1.32 % average divergence within species of Daphnia and a maximum divergence of 4.3%. In comparison, geographic clades within named species of Scapholeberis are often beyond 20% in divergence. These unusually high maximum values for Scapholeberis are unlikely to be reduced with further geographic sampling. The COI data showed similar levels of within genus variation for Daphnia (Costa et al., 1997), Scapholeberis and Megafenestra at just over 30%, while the rRNA genes show greater divergences within neustonic genera (Scapholeberis and Megafenestra) compared to those from other cladoceran genera (Taylor, Connelly & Kotov, 2020). This outcome is expected for rate increases in COI because the gene is prone to strong purifying selection resulting in substitutional saturation (Pentinsaari et al., 2016).

The COI based tree (Fig. 2) is similar to the tree estimated from 16S+12S rRNA sequences (Taylor, Connelly & Kotov, 2020). The major groups in both trees are the same, while the grouping of the deep branches is different. But, as the deep branches for COI have low support, the discrepancies may be due to random error.

The mucronata group is well-supported in both trees, in each tree the group is represented by four main clades. Our study confirms that the mucronata-group (clade X) is present in non-Beringian North America. Clade X is known only from COI sequences from Manitoba, Canada (Jeffery, Elías-Gutiérrez & Adamowicz, 2011).

All clades from the rammneri group represented in the rRNA tree (Taylor, Connelly & Kotov, 2020) are also present in the COI tree (Fig. 2). New information includes: (1) Clade H penetrates further north in the Nearctic (though not beyond the boreal zone); (2) there is a previously unknown clade Y in Israel; (3) the grouping of clade I (which is also basal in the rRNA tree) with other clades is not well-supported in the COI tree.

The present study has much improved the geographic sampling of the S. freyi group compared to our rRNA tree (this is largely due to the inclusion of sequences from previous DNA barcoding projects). It is clear from the present results that S. freyi is indeed a diverse clade with many closely related, but geographically differentiated phylogroups in the New World.

There is a new genetic clade within the S. kingii species group, S. cf. intermedius (clade L2) (Figs. 1–2) which was not sampled in the rRNA study. Therefore, the S. kingii group is more diverse as it was expected before. In our COI tree, S. armata (clade N) grouped with S. cf. microcephala (clade E) (Fig. 2), but they are distant branches on the rRNA tree. The source of the incongruence is unknown but such discrepancies are common with long branches and short internodes (see Omilian & Taylor, 2001; Bergsten, 2005).

Finally, the Megafenestra internal tree structure is different from that in rRNA tree, as the clade P is sister group of Q in the COI tree and O – in the rRNA tree.

De-coding of the DNA barcoding results

Before our study, 48 COI sequences were deposited to GenBank: DeWaard et al. (2006) (1 sequence); Richter, Olesen & Wheeler (2007) (1); Elías-Gutiérrez et al. (2008) (6); Jeffery, Elías-Gutiérrez & Adamowicz (2011) (2); Elías-Gutiérrez & León-Regagnon (2013) (3); Prosser, Martínez-Arce & Elías-Gutiérrez. (2013) (2); Yang et al. (2017) (1); (14), and 20 sequences as direct submissions, including the iBOL releases. Because the taxonomy of the Scapholeberinae is immature, identifications of the taxa by authors of these data were tentative (Fig. 21), only 30% of taxa were identified to species group accurately, while others were misidentified or identified to the genus level. In some publications, species were assigned to numbers: e.g., “sp. 1, sp. 2 and sp. 3” of Jeffery, Elías-Gutiérrez & Adamowicz (2011). Subsequently, S. duranguensis was reasonably described from Mexico (Quiroz-Vázquez & Elías-Gutiérrez, 2009) based on specific COI sequences and morphological differences from other North American taxa, but no suggestions on the diversity within the genus were made. S. yahuarcaquensis was described recently from South America (Andrade-Sossa, Buitron-Caicedo & Elías-Gutiérrez, 2020), it corresponds to our clade J4.

Assessments of species diversity based on genetics can be confused by an immature taxonomic scaffold (as in Scapholeberis and Megafenestra). Indeed, before our study, GenBank was a source of misidentification, as 70% of sequences had incorrect labels. The barcoding data were an illegitimate alternative to real taxonomy based on the species typification and accurate descriptions/identifications (see Kotov & Gololobova, 2016). Moreover, when there are pervasive rate differences among taxa for mitochondrial DNA, as has been proposed for neustonic daphniids (Taylor, Connelly & Kotov, 2020), mitochondrial DNA approaches may yield very different diversity estimates from morphological or nuclear genomic evidence.

Our recent decoding of the data from GenBank led to several interesting conclusions. The owners of sequences had no chance to make them because the barcoding data were not well-integrated with taxonomy. Note that the following conclusions are mainly based on the analysis of the GenBank sequences rather than our original data:S. freyi is not a subspecies of S. armata, and even not single monotypic species, but a monophyletic group of closely related genetic lineages (potential biological species) with a clear latitudinal differentiation in the Americas. Our previous hypothesis that S. freyi is a part of S. rammneri group (Taylor, Connelly & Kotov, 2020) was wrong. Note that to date only S. freyi s. lat. is genetically detected in tropical South and Central America. This conclusion agrees with opinions based on morphological data (Elmoor-Loureiro, 2000; Elías-Gutiérrez, Kotov & Garfias-Espejo, 2006). In contrast, S. freyi has not been detected in the western half of the Nearctic. S. yahuarcaquensis was also found in Europe—this population is most probably the result of human-mediated introduction (see also Taylor, Connelly & Kotov, 2020). The European population was used for a genomic study and identified as “S. mucronata group” (Cornetti et al., 2019).

S. duranguensis is a member of a large group, namely the S. freyi species group. It is not micro-endemic of a single locality in Durango State, but also present in the mountains of Aguascalientes State.

Members of the S. mucronata group (namely clade X) are present in non-Beringian North America, but probably only in its northernmost (Arctic) portion.

A new lineage (most probably, a separate biological species) of the rammneri group is present in Israel.

In contrast to our previous opinion (Taylor, Connelly & Kotov, 2020) representatives of the American clade H of the rammneri group are found in the Beringian zone (although they probably do not extend beyond the boreal zone in Alaska).

The information from "genetic barcoding" allows us to improve the biogeography of neustonic daphniids, but only after integrating this information with morphological and other genetic data (Schlick-Steiner et al., 2010).

Taxonomy

There are two species within the genus Megafenestra (Dumont & Pensaert, 1983): M. aurita (Fischer, 1849) and M. nasuta (Birge, 1879), and eleven valid species within the genus Scapholeberis: (1) S. mucronata (O.F. Müller, 1776); (2) S. spinifera (Nicolet, 1849); (3) S. armata Herrick, 1882; (4) S. kingii Sars, 1888; (5) S. microcephala Sars, 1890; (6) S. erinaceus Daday, 1903; (7) S. rammneri Dumont & Pensaert, 1983; (8) S. freyi Dumont & Pensaert, 1983; (9) S. duranguensis Quiroz-Vázquez & Elías-Gutiérrez, 2009; (10) S. yahuarcaquensis Andrade-Sossa, Buitron-Caicedo & Elías-Gutiérrez, 2020; (10) S. smirnovi sp.nov.

But at least four “species” from this list (S. kingi, S. microcephala, S. mucronata, S. rammneri) could be considered as taxa requiring special attention to their taxonomy due to their very broad ranges both in the Eastern and Western Hemispheres. Such taxa need careful taxonomic revisions according to the logic of “non-cosmopolitanism” and “continental endemism” approaches (Frey, 1982; Frey, 1987) widely accepted in the cladoceran taxonomy and biogeography.

After two subsequent revisions (Taylor, Connelly & Kotov, 2020; this study) we know that the diversity of both genera has been strongly underestimated. The subfamily includes at least 23–24 distinct lineages (note that rare S. erinaceus was not studied either here or by Taylor, Connelly & Kotov (2020). In contrast to many other cladoceran groups, we can confidently say that the phylogeny and taxonomy of Scapholeberinae is now relatively well-done. Main species groups correspond well to those separated based on the morphological analysis. But it is very obvious that further studies are necessary to find morphological differences between revealed taxa and formulate diagnoses of the taxa which needs to be formally described (as Megafenestra cf. nasuta clade P, Scapholeberis cf. microcephala clade E., S. cf. rammneri clades I, and possibly other un-named clades). Therefore the revision of the taxonomy only starts with this contribution. After all, from the lineages discovered up to now (and there are surely more to be found), most remain unnamed and phenotypically not characterised.

To date we do not know if these taxa are morphologically different from congeneric taxa. But, in this context, it is very premature to discuss a “lacking of resolution” of morphology and the “limitations inherent in morphology-based identification system” (Hebert et al., 2003: p. 313), as nobody tried to find such differences. Such a search is a task for the future.

We can immediately recommend the main direction of such studies: gamogenetic specimens must be analyzed for diagnostic characters first, as we did for the S. kingii species group. We can assume, following ideas of Goulden (1966), that differences in the ephippial morphology could provide a mechanism of reproductive isolation. As such differences could be cues for the male during copulation to recognize the correct mate. Lateral keels on the ephippium, characteristic of several, if not all, taxa of Scapholeberis, are analogous to the keels in Bosminidae (Kotov, 2013). Kerfoot & Peterson (1980) proposed that the lateral keels and special texture on the ephippia of Bosmina also contribute to pre-zygotic reproductive isolation. We believe that differences between Scapholeberis ephippial females could also contribute to reproductive isolation among congeneric species. Moreover, the situation with Scapholeberis kingii and S. smirnovi sp.nov., when parthenogenetic females are morphologically indistinguishable, but gamogenetic specimens have morphological differences, is usual among the cladocerans (Belyaeva & Taylor, 2009; Popova et al., 2016; Smirnov & Kotov, 2018). Such phenomena need further study to be accurately explained, but it is obvious that the morphological evolution in parthenogenetic and gamogenetic specimens follow somewhat different pathways. The oft-reported morphological stasis in cladocerans (Sacherová & Hebert, 2003; Smirnov & Kotov, 2018) is more characteristic of parthenogenetic females (the sexual stages appear to evolve more rapidly in morphology).

Supplemental Information

Supplemental Information 1 Complete list of original sequences obtained in the frame of this study and GenBank sequences with information on specimen ID and locality provided for each individual.

Click here for additional data file.

Supplemental Information 2 List of material studied morphologically.

Click here for additional data file.

Supplemental Information 3 Neighbor joining tree of neustonic daphniids (Scapholeberis and Megafenestra) using Kimura’s 2 parameter distances.

Bold letters (A–Q, X–Y) indicate geographic clades. Colours represent major species groups in the Scapholeberinae: Scapholeberis mucronata group (green), S. rammneri group (red), S. freyi group (black), S. kingii group (grey), genus Megafenestra (white). The tree is midpoint rooted supporting a basal position of the genus Megafenestra. See Appendix S1 for individual sequences.

Click here for additional data file.

Supplemental Information 4 Set of COI sequences used in the study.

Click here for additional data file.

Many thanks to E.I. Bekker, J.F. Cart, A. V. Chabovsky, S.E. Cherenkov, S.J. Connelly, Y.R. Galimov, S. Ishida, P.D. Karabanov, Y. Kobayashi, T.P. Korobkova, L.E. Savinetskaya, A.B. Savinetsky and D.V. Vasilenko for samples with the Scapholeberinae, M. Ballinger, K.S. Costanzo, N.M., Korovchinsky and A. Medeiros for assistance in sampling trips. We are deeply grateful to Sergei I. Metelev and Alexey N. Nekrasov for technical assistance during SEM investigations. SEM works are carried out at the Joint Usage Center “Instrumental Methods in Ecology” at A.N. Severtsov Institute of Ecol. Evol. of Russian Academy of Sciences.

Abbreviations for collections

DAD Permanent slides from Collectio Dadayana, the Hungarian Natural History Museum, Budapest, Hungary

MGU ML Invertebrate Collection of Moscow State University, Moscow, Russia

NIBR Collection of the National Institute of Biological Resources, Inchon, South Korea

Abbreviations in illustrations and text

I–V thoracic limbs I–V

acs accessory seta

e1–e5 endites 1–5 of thoracic limbs

ejh ejector hooks on limb I

epp epipodite

ext exopodite

IDL inner distal lobe

ODL outer distal lobe

pep preepipodite

Additional Information and Declarations

Competing Interests

Author Contributions

Field Study Permissions

DNA Deposition

Data Availability

New Species Registration

The authors declare that they have no competing interests.

Petr G. Garibian performed the experiments, prepared figures and/or tables, and approved the final draft.

Anna N. Neretina analyzed the data, prepared figures and/or tables, authored or reviewed drafts of the paper, sEM studies, and approved the final draft.

Derek J. Taylor conceived and designed the experiments, performed the experiments, analyzed the data, prepared figures and/or tables, authored or reviewed drafts of the paper, and approved the final draft.

Alexey A. Kotov conceived and designed the experiments, analyzed the data, prepared figures and/or tables, authored or reviewed drafts of the paper, and approved the final draft.

The following information was supplied relating to field study approvals (i.e., approving body and any reference numbers):

Sampling in South Korea was conducted in frames of the program of the National Institute of Biological Resources (NIBR), funded by the Ministry of Environment (MOE) of the Republic of Korea (NIBR201701201), such sampling do not require special additional permission.

Sampling in Ethiopia was conducted in frames of work of the Joint Ethiopian-Russian Biological Expedition (JERBE), with permission from the Ministry of Environment of Ethiopia to JERBE.

Sampling in non-protected water bodies of Russia do not require permissions.

The following information was supplied regarding the deposition of DNA sequences:

All original sequences are available at Genbank: MT371605–MT371659.

The following information was supplied regarding data availability:

The sequences, list of material studied morphologically, and Neighbor joining tree of neustonic daphniids (Scapholeberis and Megafenestra) using Kimura’s 2 parameter distances are available as Supplemental Files.

The material is stored in three museums (Zoological Museum of Moscow State University, Moscow = MGU; Collectio Dadayana, the Hungarian Natural History Museum, Budapest, Hungary = DAD;

collection of the National Institute of Biological Resources, Inchon, South Korea = NIBR with the following numbers:

MGU Ml 189_Scapholeberis smirnovi_Holotype_1_females_Russia (Asian)_Primorski Territory_A puddle on the road to Maloe Utinoe Lake.

MGU Ml 190_Scapholeberis smirnovi_Paratypes_58_females_Russia (Asian)_Primorski Territory_A puddle on the road to Maloe Utinoe Lake.

MGU Ml 197_Scapholeberis cf. intermedius__20_females_Ethiopia_Amhara_Bahir Dar Bay of Lake Tana, Blue Nile Basin, a small oxbow 1.

MGU Ml 198_Scapholeberis smirnovi_Paratypes_14_females_Russia (Asian)_Primorski Territory_Puddle 2 near the reservoir of Luchegorskaya power station.

MGU Ml 199_Scapholeberis smirnovi_Paratypes_15_females_Russia (Asian)_Sakhalin Area_Puddles between see and road, Puzina Peninsula.

MGU Ml 200_Scapholeberis smirnovi_Paratypes_15_females_Japan_Waka-yama_Shin-Ike, Iwade County.

MGU Ml 201_Scapholeberis smirnovi_Paratypes_15_females_Russia (Asian)_Sakhalin Area_Puddle near Susuya River, Yuzhno-Sakhalinsk.

MGU Ml 202_Scapholeberis smirnovi_Paratypes_14_females_Russia (Asian)_Primorski Territory_Lake Kirpichnoe, region of Lake Khanka.

MGU Ml 203_Scapholeberis smirnovi_Paratypes_11_females_Russia (Asian)_Primorski Territory_A roadside pool near Lake Khanka.

MGU Ml 204_Scapholeberis smirnovi_Paratypes_12_females_Russia (Asian)_Primorski Territory_A puddle on the road to Maloe Utinoe Lake.

MGU Ml 205_Scapholeberis smirnovi_Paratypes_11_females_South Korea_Jeollabuk-do_Susi Reservoir.

MGU Ml 206_Scapholeberis smirnovi_Paratypes_14_females_Russia (Asian)_Primorski Territory_A pond in the valley of Razdol’naya River.

MGU Ml 207_Scapholeberis smirnovi_Paratypes_12_females_South Korea_Gangwon-do_Dongmak reservoir.

MGU Ml 208_Scapholeberis smirnovi_Paratypes_14_females_South Korea_Jeju-do_Yeonhwa pond (with Lotus).

MGU Ml 209_Scapholeberis smirnovi_Paratypes_11_females_South Korea_Jeju-do_Pond near 31 (Phragmites + Trapa).

MGU Ml 210_Scapholeberis smirnovi_Paratypes_14_females_Russia (Asian)_Khabarovsk Territory_A puddle at edge of a Sphagnum swamp, near Gerbi.

DAD 1070/159_Scapholeberis intermedius_Lectotype_1_females_Sri Lanka__"Sri Lanka", no further information.

DAD 1070/159 (on the same slide with the lectotype)_Scapholeberis intermedius_Paralectotypes_3_females_Sri Lanka__“Sri Lanka”, no further information.

DAD 1070/155_Scapholeberis intermedius_Paralectotypes_1_females_Sri Lanka__“Sri Lanka”, no further information.

DAD 1070/156_Scapholeberis intermedius_Paralectotypes_2_females_Sri Lanka__“Sri Lanka”, no further information

DAD 1070/157_Scapholeberis intermedius_Paralectotypes_1_females_Sri Lanka__“Sri Lanka”, no further information.

DAD 1070/158_Scapholeberis intermedius_Paralectotypes_3_females_Sri Lanka__“Sri Lanka”, no further information.

DAD 1070/160_Scapholeberis intermedius_Paralectotypes_3_females_Sri Lanka__“Sri Lanka”, no further information.

n/n_Scapholeberis kingii_ ephippial & parth. females_Australia_Tasmania_Roadside pool near Lake Bantic, West Coast.

n/n_Scapholeberis kingii_ ephippial & parth. females_Australia_New South Wales_Farm Dam.

NIBRIV0000870500_Scapholeberis smirnovi_Paratypes_20_females_South Korea__Simok-ri, Jigok-myeon, Hamyang-gun, Gyeongsangnam-do.

NIBRIV0000870501_Scapholeberis smirnovi_Paratypes_20_females_South Korea__Swamp beside road near Sin Yeongri Reservoir.

The following information was supplied regarding the registration of a newly described species:

Publication LSID: urn:lsid:zoobank.org:pub:A4A3415D-857E-42E5-9103-B8D48AC60832.

Scapholeberis LSID: urn:lsid:zoobank.org:act:10509BC4-57B1-4375-9E42-54FE6CF82DBC.

S. smirnovi sp.nov. LSID: urn:lsid:zoobank.org:act:62ABBAFB-249D-453A-BB8D-E59ECB1AB2B0.

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
