# Peer review of "Partial revision of the neustonic genus Scapholeberis Schoedler, 1858 (Crustacea: Cladocera): decoding of the barcoding results"

_PeerJ, doi:10.7717/peerj.10410_

## Round 0.1 · original submission · Minor Revisions

Thank you for submitting your manuscript to PeerJ. I have sent your paper to three expert referees for their consideration. I have now received their comments back and have read through your paper carefully myself. Enclosed please find the reviews of your manuscript.

All three reviews are in general favourable and find merit in your study, suggesting that, subject to minor revisions, your paper could be suitable for publication.

I would ask you to pay special attention to the comments of Reviewer 2 regarding modifications to be implemented in Figure 11; if the authors find it pertinent, please think about providing the identification key for the Scapholeberis kingi species group. Also please implement all the corrections and address the questions raised by Reviewers 1 and 3.

Please carefully consider these suggestions, and I look forward to receiving your revision.

·

Basic reporting

The manuscript is a relevant contribution to cladoceran taxonomy. The language is in general acceptable but occasionally should be improved (see general comments, and especially the annotated manuscript). Considering the high number of figures, I suggest removing at least the last one. Occasionally, I miss references for some statements (see annotated MS).

Experimental design

Presentation of the Methods may be improved (see details in the annotated MS).

Validity of the findings

The findings are valid but considering the content of the manuscript, the title should certianly be changed to match it. The present controbution is far from an integrative revision of the genus.

Additional comments

The manuscript of Garibian et al. provides some additional genetic data for the neustonic cladoceran genus Scapholeberis (partially expanding the recently published paper on the same taxonomic group - and with overlapping authors - based on different mitochondrial markers), and detailed taxonomic treatment of one species group with redescription of S. kingii and description of the new species S. smirnovi.
It is far from an integrative revision of the genus (the title should certainly be changed, as it is misleding in the present version!) but it is a valuable contribution to the cladoceran systematics, and worthy publication after modest revisions. The paper would fit nicely to a specialized systematic journal but I understand that after the removal (hopefully temporary) of Zootaxa from the Clarivate's Journal Citation Reports, the authors seek other publication outlets.

As is typical for the Kotov group, the manuscript is strong in the detailed treatment of all aspects of cladoceran morphology (including thoracic limbs), with high-quality drawings. It also provides SEM photos (though of varying quality). However, there are various repeatedly occurring typos and expressions that can be improved, which I have corrected or commented in the annotated manuscript file. The accompanying molecular data are discussed from the systematic perspective, and are important for interpretation of the morphological data and vice versa. I do have various comments on how the methods (and partly results) are presented, mostly provided in the annotated manuscript provided as a separate file.

Introduction is rather brief but to the point. When the results of molecular analyses are discussed in the manuscript, it is not always easy to differentiate between new data and those already presented by Taylor et al., this is something the authors should highlight more thoroughly. The discussion is the section that can be most improved –some of its parts seem like rather haphazardly amassed information, which, although relevant, could be arranged more smoothly. Occasionally, I miss references to statements in the text, or feel some statements are unnecessary (these parts are commented in the annotated MS).

Taxonomic sections:
I have indicated various typos and awkward expressions (that are often repeated across multiple taxa) in the annotated manuscript.
Differential diagnoses are written without comparison with other taxa, sometimes the differences are rather mentioned elsewhere in the descriptions. If the characters described in this section are unique for the given taxon, it should be clearly stated. I would nevertheless suggest that key differences between the focal taxon and the others are specifically stated. Alternatively, a table summarizing the differences in key taxonomically relevant characters among taxa or populations of the S. kingii group in a concise manner could be helpful.

Figures:
I am not convinced the violin plots (fig. 3) are informative – these are based on pairwise divergences, so obviously their shape heavily depends on the number of sequences available for a particular clade. Considering that the sequences were not selected to be representative but summarize all the available data, the patterns are not comparable among species groups. If the authors decide to keep the plots as they are, they should interpret them more thoroughly.
In the descriptions of limbs, the setae in the same figure are often repeatedly labelled with the same numerical values (when on different part of the limb). I fully understand that the vast majority of readers interested in thoracic limb structure will not have any problems with this. Those who want to learn more and want to orient themself in the figure and match it to the text, however, may find this challenging, especially when the different parts of the limb are not labelled themselves. Consider adding extra labels to differentiate limb parts.
In SEM photos, it would help if relevant structures are indicated by arrows or symbols. The photos are of very varying quality and while I understand that is often due to quality of the original material, at least the authors should make sure the readers properly interpret them.
Fig. 18C does not have a scale for parts other than A and B
Fig. 4B seems to depict a distorted specimen. Is there a reason for that?
Scale bar is invisible in fig. 20L
Fig. 21 is in my opinion unnecessary, a little text will deliver the same information, and the paper is long even without it...

·

Basic reporting

The paper, in my opinion, is an excellent example of combining classical morphology and molecular genetics methods in taxonomical studies, which are frequently biased one way or other. It deals with ecologically important genus of littoral cladocera, and has potential to be well-cited.

Figure 11 is badly composed, drawings (which are very good by themselves) are located too closely to each other, it is hard to distinguish between them. In my opinion, such composition is rather unrespectful for the readers. I strongly suggest it should be rearranged, probably changing size of each drawing. Morphology of postabdominal seta (11 K) is fully illustrated on drawing of postabdomen (11 I), figure 11 K gives no new information and can be ommitted.

I'm not a native speaker, so I'm unable to judge English of the paper.

Experimental design

No comments

Validity of the findings

No comments

Additional comments

In my opinion, the paper can be improved if authors include identification key for the kingi-group of Scapholeberis.

·

Basic reporting

This paper is a fine example of an integrative approach, and illustrates nicely how molecular markers can guide taxonomical work. The point that taxonomic labels in Genbank and iBOL are often preliminary or even directly wrong is not new, but underlines the importance of supporting barcode libraries with voucher specimens available for morphological analysis by specialists. An interesting aspect is the observation of evolutionary rate differences between genera, which bears on the utility of molecularly defined taxa (like BINs in the iBOL database).
The paper includes COI results that complement mtRNA results from a previous paper. Although the phylogenetic reconstructions based on these markers are largely congruent, some interesting discrepancies occur. These are briefly discussed. The authors conclude that the phylogeny and taxonomy of the subfamily is now “relatively well-done”, but I suspect that this conclusion is slightly premature. A number of clades need further morphological examination, and expanding the geographic coverage is likely to refine the picture. For example, S. microcephala is not represented by populations from its terra typica. Nuclear markers may also alter the picture. Nonetheless, this work represents a significant step forward.
The sequence data are apparently deposited in Genbank as stated, but they are not released yet so I could not verify them.
I have no major objections to this work, and the following comments pertain to details. I hope they can be of some help. In my opinion only a minor revision is require.
Specific comments
Materials and methods, Genetics: References should be provided for the various software applied, as done for MrBayes. These are mostly well known, but the statement about BOOSTER providing less ‘eroded’ support values particularly needs a reference (presumably Lemione et al. 2018).
L. 102-103 «Selected individuals were placed into the plates…» What kind of plates is meant?
L. 110-111 Please provide information on primers used for sequencing
The bulk of this paper deals with morphological details, and below follow a series of details from descriptions and figures which should be addressed. I note that in figures of limb III some setae are labelled with a question mark. I recommend that the use of question marks should be explained in the Materials and methods section. There are some inconsistencies in how these setae are referred to in the text and in the figures (see below).
L. 153 OLD should be ODL
L. 194 må sjekkes NCBI
L. 317 Limb III endite 3 seta «5?» in text is probably designated just «?» in Fig. 6C.
L. 320 Seta 1 in text (on distal corner of the gnathobase) seems to be labelled “1?” in Fig. 6C
L. 470 Reference to Fig. 11 D is erroneous, as the figure shows the labrum (not the ventral carapace margin)
L. 476 Delete “in” in last sentence
L. 480 Reference to Fig 12 I-J should be Fig. 11 I-J
L. 482 ditto
L 484 Reference to Fig. 10 L should be Fig. 11 L
L. 498-499 “Two ejector hooks of different length. Two ejector hooks almost similar in size”. According to the figures, only the latter statement seems to be correct.
L. 508 Seta called “5” in text probably corresponds to “?” in Fig. 13G
L. 511 Seta called “1” in text seems to be “1?” in Fig. 13G
L. 569 Reference to Fig. 16E should be Fig. 17E
L. 571 Reference to Fig. 16E must be a mistake
L. 579 Reference to Fig. 16F should probably be Fig 16C (?)
L. 585 Reference to Fig. 17A should probably be Fig 16A (?)
L. 614 Reference to seta a-d covered by short denticles – should probably be setae 1 and 2?
L. 646 “cased a surprise” – maybe “caused a surprise”?
L. 647 Insert “the” before “real distribution”
L. 652 Insert “the” before “distribution”

Figure 1, Legend, final sentence: «…just the only localities are represent» doesn’t make sense, please rephrase
Figure 2, Legend: If colours represent species groups, please add information on the blue-coloured group (S. cf. microcephala group?)

Experimental design

The study includes all material available at the time of submission, and is in that sense comprehensive. The data are adequate for testing the phylogenetic hypotheses addressed. No objections.

Validity of the findings

The conclusions drawn are supported by the data and the analyses presented. No objections.

---

## Round 0.2 · Minor Revisions

Thank you for resubmitting your manuscript to PeerJ. I believe that the manuscript is almost ready for publication and subject to minor revisions, your paper could be suitable for publication. Please correct minor typos and address suggestions given in the attached file. I look forward to receiving the revised version of your manuscript.

·

Basic reporting

The revised manuscript have improved in most aspects that were commented upon by the referees. While I do believe that some figures may have been removed or improved, I understand the authors' decision to stick to their original ideas.

Experimental design

no comment

Validity of the findings

no comment

Additional comments

I have throroughly checked the non-systematic parts of the revised paper (and only briefly went through the tracked changes in the systematic part). I trust the authors they have carefully checked all details in the systematic section.

I do have a few additional minor corrections in the text (typos, inconsistencies, suggested improvements), these are marked in the annotated PDF. Overall, I am happy with the present version, and look forward to seeing it published.

---

## Round 0.3 · accepted · Accept

Thank you for taking the time to revise and resubmit your manuscript. I have now read through your paper as well as your letter in response to the reviews. I think that you have successfully addressed all of the concerns raised very well, and would like to accept your manuscript for publication in PeerJ. Congratulations!!!

Thank you for all the hard work you have put into this. Your paper makes a strong contribution to the literature and I look forward to seeing it published.